# Exploring the Paradox of Bone Mineral Density in Type 2 Diabetes: A Comparative Study Using Opportunistic Chest CT Texture Analysis and DXA

**DOI:** 10.3390/diagnostics13172784

**Published:** 2023-08-28

**Authors:** Min Woo Kim, Jung Wook Huh, Young Min Noh, Han Eol Seo, Dong Ha Lee

**Affiliations:** Department of Orthopedic Surgery, Busan Medical Center, 62, Yangjeong-ro, Busanjin-gu, Busan 47227, Republic of Korea; drkimminwoo@naver.com (M.W.K.); gizer00@hanmail.net (J.W.H.); doctornoh77@naver.com (Y.M.N.); mdseo86@gmail.com (H.E.S.)

**Keywords:** dual-energy X-ray absorptiometry (DXA), computed tomography texture analysis, type 2 diabetes (T2D), Bone Mineral Density (BMD)

## Abstract

Background: This study aimed to validate the application of CT texture analysis in estimating Bone Mineral Density (BMD) in patients with Type 2 Diabetes (T2D) and comparing it with the results of dual-energy X-ray absorptiometry (DXA) in a normative cohort. Methods: We analyzed a total of 510 cases (145 T2D patients and 365 normal patients) from a single institution. DXA-derived BMD and CT texture analysis-estimated BMD were compared for each participant. Additionally, we investigated the correlation among 45 different texture features within each group. Results: The correlation between CT texture analysis-estimated BMD and DXA-derived BMD in T2D patients was consistently high (0.94 or above), whether measured at L1 BMD, L1 BMC, total hip BMD, or total hip BMC. In contrast, the normative cohort showed a modest correlation, ranging from 0.66 to 0.75. Among the 45 texture features, significant differences were found in the Contrast V 64 and Contrast V 128 features in the normal group. Conclusion: In essence, our study emphasizes that the clinical assessment of bone health, particularly in T2D patients, should not merely rely on traditional measures, such as DXA BMD. Rather, it may be beneficial to incorporate other diagnostic tools, such as CT texture analysis, to better comprehend the complex interplay between various factors impacting bone health.

## 1. Introduction

The rapidly increasing prevalence of osteoporosis, distinguished by compromised bone strength and an elevated risk for fractures [1,2,3], presents a significant health concern in aging populations. In developed countries, approximately 40% of women and a proportionally lower number of men are likely to experience osteoporosis-related fractures during their lifetime [1,4,5]. Concurrently, the incidence of diabetes, a condition often coexisting with osteoporosis, is rising in the elderly population.

The global obesity epidemic contributes further to this challenge, leading to a surge in obesity-related diabetes. Currently, it affects over 366 million adults worldwide, and projections indicate a climb to 552 million by 2030 [1,2]. Intriguingly, both Type 1 and Type 2 diabetes have been linked with an elevated fracture risk, despite the paradoxical observation that individuals with Type 2 Diabetes (T2D) often exhibit higher bone mineral density (BMD) as measured by Dual-energy X-ray absorptiometry (DXA) than their non-diabetic counterparts [6,7].

As recent research increasingly identifies T2D as an independent risk factor for fractures, it is crucial to delve deeper into the associated pathophysiological mechanisms [8,9,10]. Such exploration should also aim to integrate these findings into risk assessment and treatment algorithms. This paper reviews the existing evidence on how T2D impacts bone metabolism and fracture risk, thereby highlighting the BMD paradox in T2D [11,12,13,14].

In T2D, despite a paradoxical normality in BMD, research suggests an elevated fracture risk due to potential diabetes-related skeletal alterations. These modifications could include changes in the material, dynamic, and microarchitectural properties of the bone. Nonenzymatic glycation due to high glucose levels in T2D causes an accumulation of advanced glycosylation end-products (AGEs) in the bone matrix, which can render the bone brittle and prone to fracture [9,14,15]. AGEs could also interfere with osteoblast development and function, contributing to decreased bone formation. However, the exact impact of T2D on bone formation remains to be definitively established.

In clinical practice, DXA remains the primary tool for assessing BMD and bone mineral content (BMC) [16]. However, the assessment using DXA often becomes intricate due to factors such as the progression of osteosclerosis and the degree of adiposity, and this complexity is amplified when assessing the lumbar spine BMD and BMC compared to the femoral neck bone density [17]. Recognizing these limitations, the Fracture Risk Assessment Tool (FRAX) plus model has recently been introduced, uniquely incorporating T2D and falls alongside other fracture risk factors. The World Health Organization’s traditional FRAX algorithm, which integrates several validated fracture risk factors independent of BMD, remains popular, especially in western countries. However, in scenarios where BMD data are absent, BMI is recommended as a surrogate due to its similar risk profile. Despite Type 1 diabetes being acknowledged as a secondary osteoporosis cause in the FRAX model, the exclusion of Type 2 diabetes (T2D) has raised questions about its comprehensive accuracy. Our investigations highlight that T2D patients typically demonstrate lower projected major osteoporotic fractures (MOF) risks over the subsequent decade, especially for those with a BMI below 28 kg/m^2^. Such findings underscore the potential of the FRAX tool to underestimate fracture risks in T2D patients, suggesting that the FRAX-BMD model is potentially more appropriate for assessing hip fracture risks in this demographic [18].

In this pioneering study, we propose a novel approach using axial cuts from chest CT scans at the L1. Our aim was to obtain a comprehensive understanding of bone health by investigating the correlation between BMD, BMC, and texture analysis values derived from these CT scans. This innovative method offers a more efficient way of evaluating bone mineral status, especially beneficial for patients who do not meet the criteria for DXA.

By analyzing Hounsfield unit (HU) values in CT and extracting model-based texture features via a gray-level co-occurrence matrix (GLCM), we establish linear regression models informed by DXA measures. Our cross-sectional study underscores the potential of texture analysis as a promising tool for monitoring bone mineral status, contributing to more accurate osteoporosis diagnosis and management, and providing insights into the BMD paradox seen in Type 2 diabetes patients.

## 2. Materials and Methods

### 2.1. Study Population and Selection Criteria

Our institutional review board-approved study (P01-202109-21-014) initially included a pool of 1693 patients (767 males and 926 females). All individuals underwent chest CT and DXA scans at our institution between 9 May 2011 and 30 June 2022. We then narrowed down this initial cohort to 510 patients (214 males and 296 females) based on specific selection criteria: (1) a temporal gap of less than a month between the CT and DXA scan dates, and (2) the presence of actual measurable L1 axial cuts in the CT images.

In the subsequent selection phase, we excluded patients meeting any of the following criteria: (1) the absence of a measurable axial cut of the L1 vertebra in the CT images; (2) historical evidence of L1 compression, burst fractures; (3) a history of surgical interventions due to fractures, including vertebroplasty or kyphoplasty for L1 compression fractures, or the presence of metal artifacts from unstable burst fractures; (4) challenges in identifying trabecular bones due to severe osteolytic or pathological changes.

This selection process led to the exclusion of 1183 patients, leaving a final cohort of 510 patients for inclusion in the analysis (Figure 1). This rigorous selection process was implemented to ensure the reliability of our study, concentrating on the most pertinent cases for investigating the potential of texture analysis for monitoring bone mineral status.

### 2.2. CT and DXA Scanning Procedures

Our study utilized a Siemens scanner (SOMATOM 128, Definition AS+; Siemens Healthcare, Forchheim, Germany) to conduct CT scans strictly following a standardized protocol [19,20]. Single-energy CT scans were set at 120 kVp and 247 mA parameters, with a collimation dose modulation of 0.6 mm. A consistent effective pitch of 0.8 was maintained, and the B60 (sharp) reconstruction kernel was applied. Non-contrast chest CT scans were performed, preserving a reconstructed slice thickness of 5.0 mm.

We conducted DXA scans using a standardized device following a common protocol (GE Lunar Prodigy, GE Healthcare, Freiburg, Germany). The resulting reports were compiled with vendor-specific software (Physicians Report Writer DX; Hologic, Discovery, WI, Mississauga, USA). This rigorous observance of established imaging protocols ensured the validity and uniformity of our results.

### 2.3. Defining Regions of Interest

For accurate statistical evaluations, the regions of interest (ROIs) were limited to the trabecular part of the bone, minimizing measurement distortions. Of the various ROI isolation methods, we chose the thresholding method for this research [21].

In our study, a 2-dimensional (2D) slice image from each patient’s CT axial cut was selected, specifically because it encompassed the maximum axial trabecular area of the L1 spine. As displayed in Figure 2, texture analysis was performed within a circular region that contained most of the trabecular area. It is worth noting that AGEs, a group of compounds linked to aging and diabetes, may affect the accuracy of BMD measurements [11,12,22]. The accumulation of AGEs within bone collagen can alter the mechanical properties of bone, potentially skewing the results of imaging-based assessments, such as ours.

### 2.4. Feature Derivation and Analysis

For this study, a total of 45 features were extracted from the defined regions of interest (ROIs)—five of these were intensity-based features retrieved through a histogram, and the remaining 40 were texture-based features gathered from a GLCM. These features were subsequently incorporated into two predictive models: a Linear Regression (LR) model and an Artificial Neural Network (ANN) model. The LR model estimated BMC & BMD through a linear combination of 45 features. In contrast, the ANN model was a fully interconnected network comprising three hidden layers, each containing eight, eight, and two nodes. Each node incorporated a rectified linear unit, a non-linear operator.

The intensity-based features—mean, standard deviation, skewness, kurtosis, and entropy—were extracted using an ROI image histogram [23]. These metrics provide insights into several characteristics of bone intensities, such as brightness, asymmetry, randomness, uniformity, and sharpness. On the other hand, texture analysis yielded 40 additional features that captured the spatial relationships between neighboring pixels in a 2D image. Derived from the GLCM, these texture features provide information on the characteristics of the image texture, such as entropy, contrast, correlation, homogeneity, and variance [23].

### 2.5. Correlation Examination

To estimate BMC and BMD and examine the correlation between these estimates and DXA BMD reference values, we employed either an LR model or a fully interconnected ANN. The values of the features and BMD references for each sample (case) were normalized for pre-processing. The LR model estimated BMD as a weighted combination of 45 features and a single bias term, symbolized as:yj = w0 + ∑wi * xij

The weights (wi) in this equation were optimized to minimize the mean squared error (MSE) between predicted and actual BMD values, denoted as:e = ∑ (yj − yj)^2^

This analysis was performed using a pseudoinverse in the form of a normal equation. As our sample set exceeded the number of trainable weights, we did not need to be concerned with potential overfitting issues. Consequently, we did not divide the set into training and testing subsets, nor did we implement any regularization methods, such as ridge regression or least absolute shrinkage and selection operator (LASSO) [24,25,26,27] (Table 1).

### 2.6. p-Value of the 45 Texture Feature Analysis between the Two Groups 

In our research, we scrutinized all 45 features to find those exhibiting statistically meaningful correlation differences between the T2D and normal groups. We conducted this examination across several measures, including L1 BMD, L1 BMC, Total hip BMD, and Total hip BMC.

We compiled eight distinct statistical graphs to capture the diverse perspectives provided by the data. By analyzing these graphs, we identified shared characteristics and differences in various aspects of bone data between the two groups. These observations contribute to a more comprehensive understanding of the correlation between bone texture and BMD. They also help highlight the features that indicate significant variations between the T2D and normal groups. Our findings open potential pathways for improved diagnosis and treatment strategies for bone-related diseases by pointing out the crucial features that could serve as key markers of bone health.

## 3. Results

### Patient Demographics

Our study included a total of 510 instances involving 145 T2D patients and 365 normal patients. The gender distribution was 65 men and 80 women for the T2D group, and 171 men and 194 women for the normal group. The average age of the T2D patients was 59.52 ± 11.26 years, while it was slightly lower at 57.12 ± 12.46 years for the normal group. Regarding Body Mass Index (BMI), the T2D patients had an average BMI of 24.79 ± 3.35 kg/m^2^, which was higher than the normal group’s average of 23.15 ± 4.85 kg/m^2^. As for the time intervals between the CT and DXA scans, the T2D patients had an average time of 1.57 ± 3.82 days, and the normal patients had an average time of 1.38 ± 4.17 days. A comprehensive overview of the patient demographics is presented in Table 2.

We aimed to provide a more nuanced understanding of BMD in the context of T2D. We investigated a comprehensive set of features from axial CT scans and performed a correlation analysis between these features and BMD measurements from DXA scans in both the T2D group and the normal group. Interestingly, in the T2D group, we observed a notably high correlation between the estimated values from our models and the actual BMD measurements. Specifically, we found a correlation coefficient of 0.983 (MSE: 0.021) for L1 BMD, 0.943 (MSE: 0.712) for L1 BMC, 0.952 (MSE: 0.029) for Total-Hip BMD, and 0.934 (MSE: 0.191) for Total-Hip BMC. On the other hand, in the normal group, the correlations were comparatively moderate, presenting a correlation coefficient of 0.738 (MSE: 0.072) for L1 BMD, 0.666 (MSE: 1.766) for L1 BMC, 0.758 (MSE: 0.070) for Total-Hip BMD, and 0.699 (MSE: 0.442) for Total-Hip BMC (Figure 3 and Figure 4).

These findings provide compelling evidence for the paradoxical nature of BMD in T2D patients, who, despite having higher BMD values, experience a higher risk of fractures [2,11,22,28]. The greater correlation in the T2D group may indicate that there are factors present in T2D that are not accounted for in normal individuals that enhance the precision of our models’ estimates. This implies that the complex metabolic alterations associated with T2D may have a substantial impact on BMD.

Figure 5, Figure 6, Figure 7 and Figure 8 represent comparative graphs showing statistically significant differences between the two groups, T2D and normal, across a variety of selected features. These features, chosen from the 45 initial ones, included Constant, Mean, Standard, Skewness, Kurtosis, Entropy, Contrast H and V, Correlation H and V, Energy H and V, Homogeneity H and V, and Auto Co H and V. The variations in the estimated BMD values using these features were assessed for statistical significance.

Excluding L1 BMD, the remaining cases—L1 BMC, Total Hip BMD, and Total Hip BMC—revealed statistically significant differences predominantly in the normal group, not the T2D group. This highlights that these features’ impacts on BMD and BMC may not hold the same significance in the context of T2D. The specific features showing statistically significant variation in BMD estimation could shed light on the differing bone health dynamics between healthy individuals and those with T2D.

This intriguing finding emphasizes that understanding BMD in T2D requires a comprehensive examination of multiple factors, not merely those typically associated with bone health. It further underlines the paradoxical nature of bone health in T2D patients and provides additional insights into the underlying mechanisms that may explain the increased fracture risk despite higher BMD values in T2D patients. 

## 4. Discussion

In pursuit of improving the accuracy and effectiveness of osteoporosis detection, we applied machine learning techniques and a straightforward LR model to our texture analysis of CT HU. By constructing reliable linear regression models to estimate BMC and BMD, we managed to highlight key features in both T2D and normal groups, signifying a considerable leap forward in medical imaging research.

To validate our conclusions, we turned our attention to the pronounced correlation between our CT HU-based estimates and the actual measurements derived from DXA. The compelling correlation between BMC and BMD estimates from the L1 axial cuts and hip regions, and their corresponding DXA measurements are clearly depicted in Figure 3 and Figure 4. When these regions of interest (ROIs) aligned with their DXA measurement counterparts, we registered the maximum correlation. 

In our endeavor to understand bone health complexities in T2D, we focused on the strikingly high correlation between our model estimates and actual BMD measurements within the T2D group. Notably, our results demonstrated high correlation coefficients for L1 BMD (0.983, MSE: 0.021), L1 BMC (0.943, MSE: 0.712), Total-Hip BMD (0.952, MSE: 0.029), and Total-Hip BMC (0.934, MSE: 0.191). In contrast, the correlations were somewhat lower in the normal group, with coefficients of 0.738 (MSE: 0.072) for L1 BMD, 0.666 (MSE: 1.766) for L1 BMC, 0.758 (MSE: 0.070) for Total-Hip BMD, and 0.699 (MSE: 0.442) for Total-Hip BMC. These outcomes, depicted in Figure 3 and Figure 4, highlight our methodology’s potential to discern bone health intricacies in T2D. 

The paradoxical relationship between BMD and fracture risk in T2D patients becomes even more intriguing when considering the results of our study [29,30,31,32,33,34]. This result potentially illuminates an aspect of the paradox. AGEs, known to be elevated in T2D, may negatively influence bone health. In line with this, a study using diet-induced obesity (DIO) mice, with model obese individuals with associated pre-diabetic hyperglycemia, revealed delayed bone healing characterized by aberrant fibrillar collagen structure, including sparse and disorganized collagen fibers. These structural abnormalities were partially attributed to the accumulation of AGEs, leading to increased collagen–fiber crosslink density. Thus, despite increased BMD, the quality of the bone might be compromised due to AGEs’ detrimental effects on osteoblast development and function, potentially explaining the increased fracture risk [6,22,35,36,37].

Our findings underscore the need for a more nuanced understanding of bone health in T2D patients. While BMD, as measured by DXA, is a key factor, the texture of the bone, as evaluated by CT analysis, might provide additional insight. Given the higher correlation between these two methods in the T2D group, CT texture analysis may hold promise for a more accurate assessment of fracture risk in these patients.

Building on these results, this study shows potential avenues for the application of machine learning in the field of medical imaging. By employing traditional radiomics steps, including pre-processing, manual segmentation, and feature extraction, we were able to predict BMD. This implies that CT HU texture analysis could serve as an effective tool for BMC estimation, offering a promising alternative to conventional DXA imaging [16,38].

In this study, we harnessed the power of machine learning, specifically the ANN and straightforward LR models, to enhance the precision and efficiency of our analysis. The rapid advances in computing power have made machine learning a transformative force in various fields [39,40,41,42,43,44], including medical imaging, where it significantly enhances diagnostic accuracy. A subset of machine learning, radiomics, has grown exponentially due to its capacity to extract quantifiable features from ROIs in images. These features play a crucial role in achieving prognostic or predictive objectives [21,45]. In this study, we utilized conventional radiomics steps, such as pre-processing, manual segmentation, and feature extraction, to predict BMD [21]. The features we focused on included energy, kurtosis, and skewness from intensity, as well as texture analysis employing the GLCM.

GLCM has become widely adopted due to its ability to extract heterogeneous tissue features by considering the frequency of pixel pairs occurring together in a particular relationship [23,46,47]. Functions derived from GLCM encapsulate a range of texture properties, such as energy, contrast, entropy, autocorrelation, correlation, inverse moment, and cluster shade [23,48,49,50,51]. The integration of machine learning with radiomics has seen recent adoption, especially in Magnetic Resonance Imaging (MRI) studies. For instance, researchers have used GLCM textures and logistic regression to differentiate specific types of brain tumors. Others have expanded their feature set with methods such as Gray Level Run Length Matrices (GLRLM) and wavelet transforms [48]. By using low-dimensional handcrafted features from high-dimensional images as inputs to the deep learning model, it is possible to simplify the model structure, reducing the risk of overfitting when sample image numbers are limited.

In the context of the regression analysis shown in Figure 5, Figure 6, Figure 7 and Figure 8, the *p*-value is commonly used to test the null hypothesis that a particular coefficient in the model (e.g., the coefficient for a certain feature) is equal to zero, which would imply that the feature has no effect on the predicted variable.

In the first graph (Hip_BMC by normal), five key features, including Homo_H_128, Homo_H_64, and others, yield *p*-values below 0.05, indicating their statistical significance as predictors of hip BMC. This suggests a compelling association between these features and hip BMC, which could potentially serve as reliable indicators. In contrast, features with *p*-values exceeding 0.05 did not exhibit a statistically significant relationship with hip BMC in our model. Therefore, despite their presence, we could not establish a robust link between these features and hip BMC, suggesting their limited impact on BMC or BMD estimations.

However, it is important to note that a larger *p*-value does not necessarily mean that a feature is irrelevant—it might still have an effect, but the observed data did not provide enough evidence to confidently say so. In some cases, these features might still be useful in predicting the target variable when combined with other features, despite not being significant on their own. The process of comparing *p*-values can help prioritize which features are most valuable in predicting outcomes and can provide a deeper understanding of the significant determinants influencing these outcomes.

Our study underscores the potential of CT HU texture analysis in predicting BMC and BMD, demonstrating promising results in both T2D and normal populations. However, it is important to acknowledge certain limitations. The scope of our research was narrowed by not considering osteoporosis risk factors or the influence of specific medications, both of which could significantly affect BMC and BMD outcomes. The predictive model primarily focused on the L1 axial cut from opportunistic chest CT scans, which restricted the breadth of our findings. Moreover, the comparison with DXA measurements was limited to the L1 value rather than the standard L1–L4 range. Comorbidities in the T2D and normal groups were not taken into account, and the bulk of our data was sourced from a single institution, possibly influencing the generalizability of our results.

Despite certain limitations, our study sheds light on a novel approach to estimating both BMC and BMD, providing a foundation for future research to refine and broaden this technique. Subsequent studies employing larger and more diverse datasets, coupled with extended follow-up periods, could substantiate, and augment the utility of CT HU texture analysis in the realm of osteoporosis detection and management.

## 5. Conclusions

In essence, our study emphasizes that the clinical assessment of bone health, particularly in T2D patients, should not merely rely on traditional measures, such as DXA BMD. Rather, it may be beneficial to incorporate other diagnostic tools, such as CT texture analysis, to better comprehend the complex interplay between various factors impacting bone health.

## Figures and Tables

**Figure 1 diagnostics-13-02784-f001:**
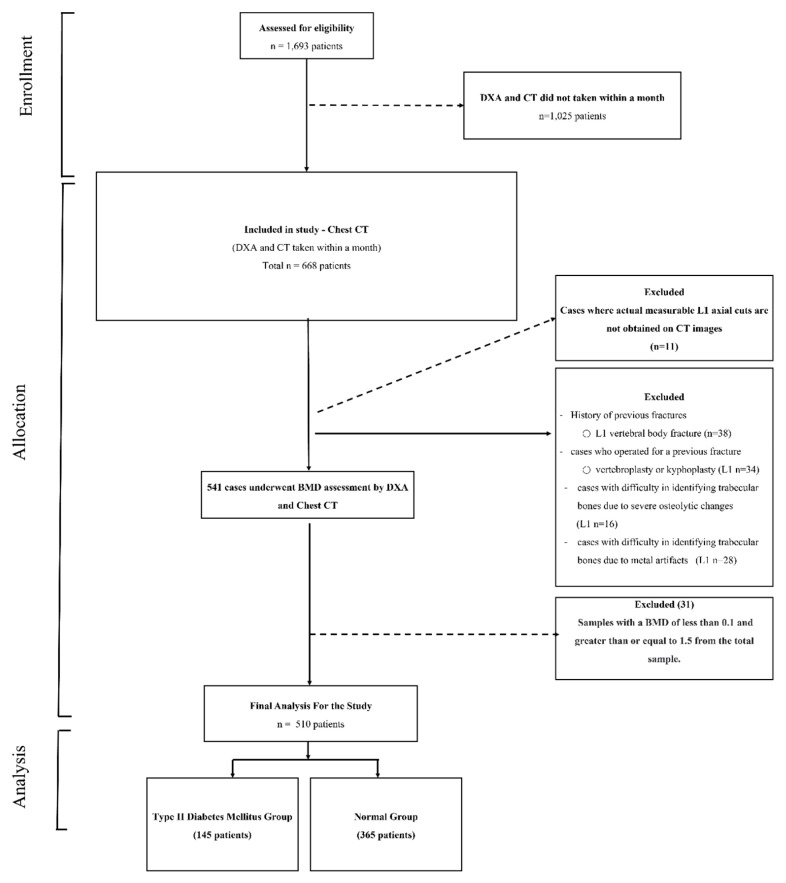
Flowchart Illustrating the Selection Process of T2D and Normal Groups.

**Figure 2 diagnostics-13-02784-f002:**
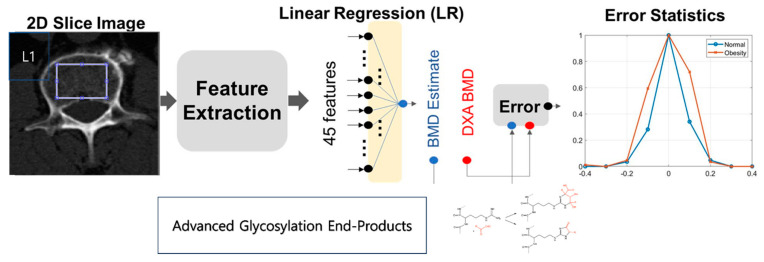
Diagrammatic Representation of BMC and BMD Estimations from Chest Computed Tomography ROI L1, Highlighting the Influence of Advanced Glycosylation End-Products.

**Figure 3 diagnostics-13-02784-f003:**
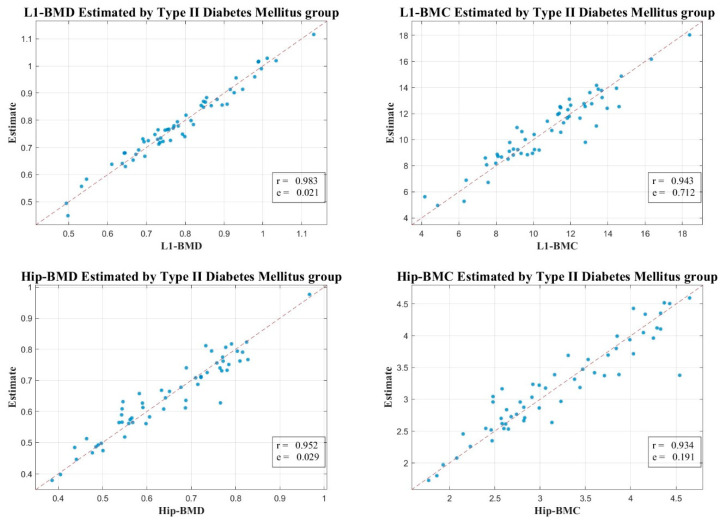
Scatter Plot Depicting the Correlation Between Estimated and Actual BMC and BMD Measurements from L1 Axial Cuts in CT and DXA, Respectively, in the T2D Group.

**Figure 4 diagnostics-13-02784-f004:**
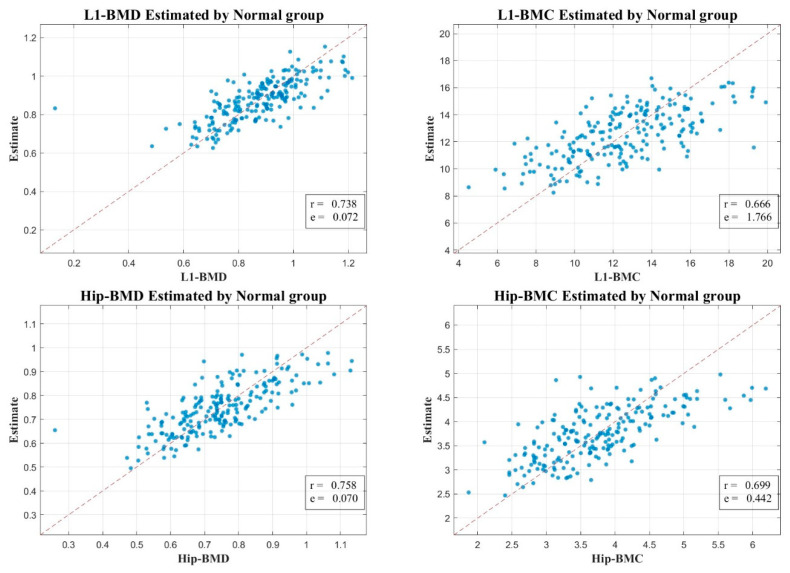
Scatter Plot Illustrating the Correlation Between Estimated and Actual BMC and BMD Measurements from L1 Axial Cuts in CT and DXA, Respectively, in the Normal Group.

**Figure 5 diagnostics-13-02784-f005:**
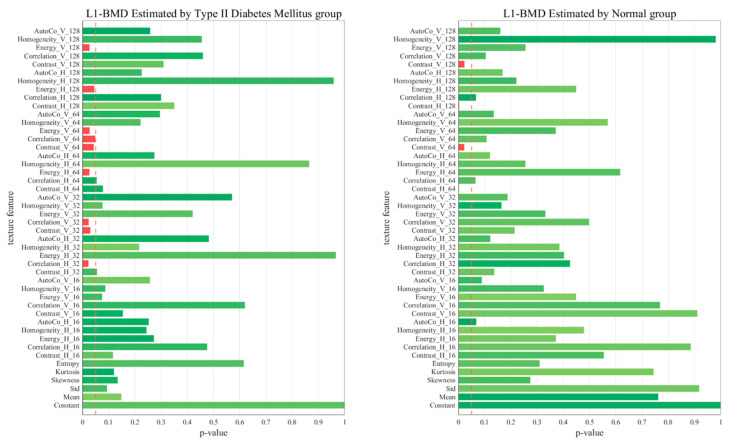
Comparative Graphs Illustrating Statistically Significant Differences in Selected Features Between the T2D and Normal Groups, as Indicated by L1 BMD *p*-values.

**Figure 6 diagnostics-13-02784-f006:**
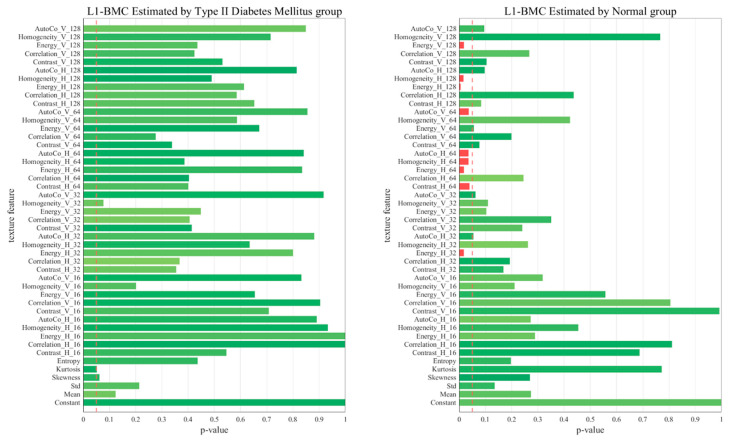
Comparative Graphs Illustrating Statistically Significant Differences in Selected Features Between the T2D and Normal Groups, as Indicated by L1 BMC *p*-values.

**Figure 7 diagnostics-13-02784-f007:**
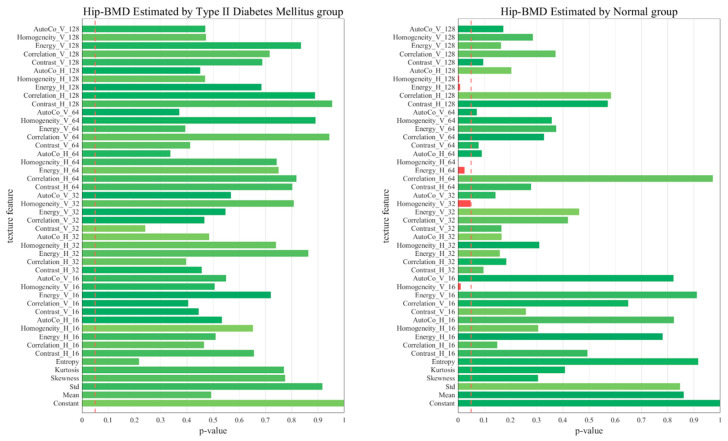
Comparative Graphs Illustrating Statistically Significant Differences in Selected Features Between the T2D and Normal Groups, as Indicated by Total Hip BMD *p*-values.

**Figure 8 diagnostics-13-02784-f008:**
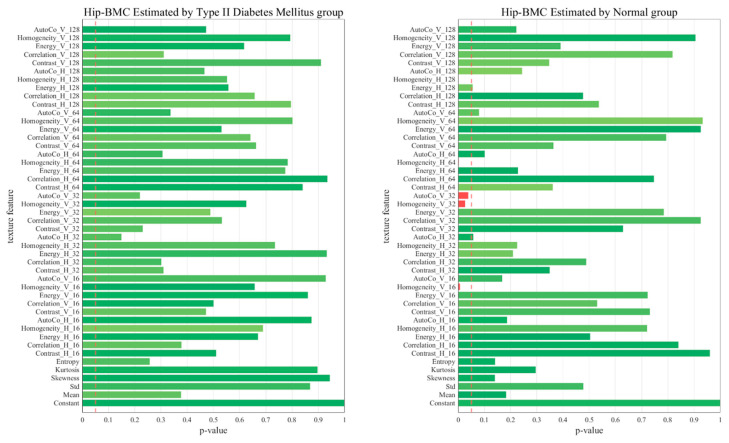
Comparative Graphs Illustrating Statistically Significant Differences in Selected Features Between the T2D and Normal Groups, as Indicated by Total Hip BMC *p*-values.

**Table 1 diagnostics-13-02784-t001:** Parameters of Gray-Level Co-Occurrence Matrix Features.

Analytical Tool	Parameter	Value/Name/Function	Feature #
Histogram	Statistics (k)	mean (k = 1), standard deviation (k = 2), skewness (k = 3), kurtosis (k = 4) entropy (k = 5)	5
Texture (GLCM)	Directions (l)	horizontal (l = 1), vertical (l = 2)	2 × 4 × 5 = 40
	Levels (m)	16 (m = 1), 32 (m = 2), 64 (m = 3), 128 (m = 4)	
	Statistics (n)	contrast (n = 1), correlation (n = 2), energy (n = 3), homogeneity (n = 4), variance (n = 5)	

**Table 2 diagnostics-13-02784-t002:** Demographic Information of Study Participants.

Case (number) (T2D/Normal)	510 (145/365)
Mean age (years) for T2D	59.52 ± 11.26
The time between CT and DXA dates (days) for T2D	1.57 ± 3.82
Sex (male/female) for T2D	65/80
BMI (kg/m^2^) for T2D	24.79 ± 3.35
Mean age (years) for normal	57.12 ± 12.46
The time between CT and DXA dates (days) for normal	1.38 ± 4.17
Sex (male/female) for normal	171/194
BMI (kg/m^2^) for normal	23.15 ± 4.85

## Data Availability

The datasets generated and/or analysed during the current study are not publicly available because we did not obtain authorization from the patients for disclosure regarding patient privacy. However, datasets are available from the corresponding author on reasonable request.

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
