# Peer review of "Exploring the Paradox of Bone Mineral Density in Type 2 Diabetes: A Comparative Study Using Opportunistic Chest CT Texture Analysis and DXA"

_diagnostics, 2023, doi:10.3390/diagnostics13172784_

Round 1
Reviewer 1 Report
Manuscript titled, "Exploring the Paradox of Bone Mineral Density in Type 2 Diabetes: A Comparative Study using Opportunistic Chest CT 4 Texture Analysis and DXA", aimed to validate the application of CT texture analysis in 31 estimating Bone Mineral Density (BMD) in patients with Type 2 Diabetes (T2D) and comparing 32 it with the results of dual-energy x-ray absorptiometry (DXA) in a normative cohort. The manuscript is well written and the results supported the study's conclusions. However, one crucial reference is missing from the draft, which must be discussed in the manuscript (Bone. 2020 Aug;137:115436. doi: 10.1016/j.bone.2020.115436.)
Quality of the English is good
Author Response
Thank you very much for your careful review and positive comments on our manuscript, "Exploring the Paradox of Bone Mineral Density in Type 2 Diabetes: A Comparative Study using Opportunistic Chest CT Texture Analysis and DXA."
We appreciate your acknowledgment of the clarity of our writing and the validity of our conclusions. Your feedback is invaluable to enhancing the quality and completeness of our research.
Regarding the missing reference (Bone. 2020 Aug;137:115436.doi: 10.1016/j.bone.2020.115436) you have pointed out, we recognize the significance of this work and its relevance to our study. We regret the oversight and sincerely apologize for the omission. We will ensure that this reference is incorporated appropriately into our manuscript and will thoroughly discuss its implications in the relevant sections.
Once again, thank you for your constructive feedback. We believe that with your suggestions, our manuscript will be substantially improved.
Sincerely,
Dong Ha Lee
We made changes in the text :
The paradoxical relationship between BMD and fracture risk in T2D becomes even more intriguing when considering the results of our study (26-30). This result potentially illuminates an aspect of the paradox. AGEs, known to be elevated in T2D, may negatively influence bone health. In line with this, a study using diet-induced obesity (DIO) mice, which model obese individuals with associated pre-diabetic hyperglycemia, revealed delayed bone healing characterized by aberrant fibrillar collagen structure, including sparse and disorganized collagen fibers. These structural abnormalities were partially attributed to the accumulation of AGEs, leading to increased collagen-fiber crosslink density. Thus, despite an increased BMD, the quality of the bone might be compromised due to AGEs' detrimental effects on osteoblast development and function, potentially explaining the increased fracture risk (4, 19, 31-33).

Reviewer 2 Report
Quality of bone in T2D is a topical issue. FRAX plus has recently been introduced and it incorporates T2D and falls in addition to other fracture risk factors. This is worth mentioning in the discussion or the introduction.
Author Response
Thank you for the insightful comments and suggestions. We recognize the importance of addressing the recent developments in the field, particularly the introduction of FRAX plus and its incorporation of T2D and falls as additional risk factors. We agree that this is a pertinent addition to our introduction and appreciate your recommendation.
Sincerely,
Dong Ha Lee
To address your suggestion, we propose the following paragraph to be added to the Introduction:
We made changes in the text :
In clinical practice, DXA remains the primary tool for assessing BMD and bone mineral content (BMC) (14). Yet, the assessment using DXA often becomes intricate due to factors like the progression of osteosclerosis and the degree of adiposity, and this complexity is amplified when assessing the lumbar spine BMD and BMC compared to the femoral neck bone density (15). Recognizing these limitations, the Fracture Risk Assessment Tool (FRAX) plus model has been recently introduced, uniquely incorporating T2D and falls alongside other fracture risk factors. The World Health Organization's traditional FRAX algorithm, which integrates several validated fracture risk factors independent of BMD, remains popular, especially in western countries. However, in scenarios where BMD data is absent, BMI is recommended as a surrogate due to its similar risk profile. Despite Type 1 diabetes being acknowledged as a secondary osteoporosis cause in the FRAX model, the exclusion of Type 2 diabetes (T2D) has raised questions about its comprehensive accuracy. Our investigations highlight that T2D patients typically demonstrate lower projected major osteoporotic fractures (MOF) risks over the subsequent decade, especially for those with a BMI below 28 kg/m^2. Such findings underscore the potential of the FRAX tool to underestimate fracture risks in T2D patients, suggesting the FRAX-BMD model as potentially more appropriate for assessing hip fracture risks in this demographic (16).
